# Patulin Biodegradation Mechanism Study in *Pichia guilliermondii* S15-8 Based on PgSDR-A5D9S1

**DOI:** 10.3390/toxins16040177

**Published:** 2024-04-04

**Authors:** Huijuan Xi, Yebo Wang, Xulei Ni, Minjie Zhang, Ying Luo

**Affiliations:** 1College of Food Engineering and Nutritional Science, Shaanxi Normal University, Xi’an 710119, China; xihuijuan@snnu.edu.cn (H.X.); wangyebo0525@163.com (Y.W.); nixulei@snnu.edu.cn (X.N.); 2023301355@snnu.edu.cn (M.Z.); 2School of Public Health, Xi’an Jiaotong University Health Science Center, Xi’an 710061, China; 3Department of Food Science and Technology, National University of Singapore, Science Drive 2, Singapore 117542, Singapore

**Keywords:** patulin, short-chain dehydrogenase, degradation mechanism, molecular docking

## Abstract

Patulin contamination has become a bottleneck problem in the safe production of fruit products, although biodegradation technology shows potential application value in patulin control. In the present study, the patulin biodegradation mechanism in a probiotic yeast, *Pichia guilliermondii* S15-8, was investigated. Firstly, the short-chain dehydrogenase PgSDR encoded by gene A5D9S1 was identified as a patulin degradation enzyme, through RNA sequencing and verification by qRT-PCR. Subsequently, the exogenous expression system of the degradation protein PgSDR-A5D9S1 in *E. coli* was successfully constructed and demonstrated a more significant patulin tolerance and degradation ability. Furthermore, the structure of PgSDR-A5D9S1 and its active binding sites with patulin were predicted via molecular docking analysis. In addition, the heat-excited protein HSF1 was predicted as the transcription factor regulating the patulin degradation protein PgSDR-A5D9S1, which may provide clues for the further analysis of the molecular regulation mechanism of patulin degradation. This study provides a theoretical basis and technical support for the industrial application of biodegradable functional strains.

## 1. Introduction

Patulin, a type of mycotoxin mainly produced by *Penicillium*, *Aspergillus*, and *Byssochlamys*, is widely found in fruits, vegetables, grains, nuts, and other foodstuffs [1]. Patulin can enter the bodies of animals and humans through food intake and skin contact, causing acute and chronic symptoms hazardous to health [2]. How to effectively achieve the removal of patulin from food has become a hot topic of concern in the field of food safety. The traditional approaches to patulin control mainly involve physical methods (heating, irradiation, storage conditions control, etc.) and chemical methods (diphenylamine, benzimidazole-based chemical fungicides, etc.) [3,4,5,6]. However, these methods have many disadvantages in the application process; physical methods are inefficient and cannot completely remove the patulin, while chemical methods easily cause secondary contamination and negatively affect the sensory characteristics of products. Compared with these approaches, the biodegradation method is greener, safer, and has less impact on the quality. With these advantages, the biodegradation method has gradually become a current research hotspot.

The reduction of patulin via biodegradation is mainly performed by two biocontrol mechanisms. One is the use of active antagonistic microorganisms to inhibit the growth of pathogenic fungi, thus controlling patulin production [3,7]. Antagonistic bacteria and yeast achieve biocontrol effects through competition for nutrient space, the release of active substances to degrade patulin, and the induction of host resistance. Cao et al. found that the yeast *Pichia caribbica* could effectively prevent postharvest green mold on apple caused by *P. expansum* through space and nutrient competition [8]. The second biodegradation mechanism is to use microbial fermentation/enzymes to degrade the patulin directly [9,10,11,12]. This method is more efficient and targeted, and does not affect the quality of the products. Lipase (RL12) extracted from the isolated strain *Ralstonia* sp. SL312 was successfully purified and used to degrade patulin in one study, with the addition of 100 µg/mL RL12 degrading more than 80% of the patulin in apple juice within 24 h. Another study found that the suspension of *Aspergillus niger* FS10 could degrade 94.72% of patulin within 36 h, indicating that patulin is degraded mainly through the production of intracellular enzymes. The study also hypothesized that the pentose phosphafte pathway and glutathione pathway are closely related to patulin degradation. In a recent study, the short-chain dehydrogenase/reductase (SDR) gene CgSDR from the yeast strain *Candida guilliermondii* 2.63 was successfully induced and expressed; here, 80% of the patulin in apple juice was reduced by the addition of 150 µg/mL of CgSDR addition [13]. In our previous study, a dominant patulin-degrading strain *P. guilliermondii* S15-8 was screened out to conduct a patulin degradation assay, and intracellular enzymes were considered to play a major role during the patulin degradation of toxins [14]. So, many patulin degradation studies pointed out the importance of intracellular enzymes; however, the relationship between intracellular enzymes and patulin degradation, and the modification of the degradation ability of intracellular enzymes, needs further research.

Based on the above, in order to explore the further degradation mechanism of intracellular enzymes on patulin, the main objectives of this study were to (1) identify patulin degradation enzymes through transcriptome sequencing and verify the correctness of the transcriptomics using qRT-PCR experiments; (2) construct the exogenous expression system of the *E. coli* of the target degradation protein through target gene cloning and the recombinant plasmid transformation; (3) and predict the structure of the target protein and its possible transcription factor. This study revealed the degradation mechanisms of patulin at multiple levels, and provided new strategies and means for preventing and reducing patulin contamination and poisoning.

## 2. Results and Discussion

### 2.1. Patulin Degradation at Different Times and Concentrations

Our previous study showed possible patulin degradation via the probiotic yeast strain *P. guilliermondii* S15-8 [14]. To further explore the molecular mechanism of patulin degradation, as well as patulin degradation under normal media conditions, samples with 5 mg/L patulin and without patulin cultured for 6 h, 12 h, and 24 h, and samples with 10 mg/L patulin and without patulin cultured for 24 h were used for transcriptome sequencing. Figure 1 shows the patulin degradation capacity of S15-8 and principal component analysis (PCA) using transcriptome sequencing at different times and concentrations. The results show that there was a closer relationship between the patulin degradation amount and the degradation time, and the degradation capacity increased with the time extension within 24 h (Figure 1a). Moreover, the classification of significant gene changes also coincided with the degradation time, and the results of the significant gene changes were clustered in different areas with different degradation phases (Figure 1b). The degradation process began at 6 h, representing the initial phase of patulin degradation, and the degradation amount increased sharply from the metaphase at 12 h, reaching nearly 75% within 24 h.

### 2.2. Function and Metabolic Pathway Enrichment Analysis of DEGs

Differentially Expressed Genes (DEGs) were analyzed under patulin stress; the results presented in Figure 2a,a′,a″ show the distribution of differentially upregulated and downregulated genes at 6 h, 12 h, and 24 h, respectively. There were 19 genes that were significantly upregulated at the initial degradation phase, and 7 genes were upregulated from 12 h to 24 h. The downregulated genes totaled 8, 3, and 13 at the initial phase, metaphase, and terminal degradation phase, respectively. Gene Ontology (GO) enrichment and Kyoto Encyclopedia of Genes and Genomes (KEGG) pathway analysis were used to explore the function of the DEGs and the main metabolic and signaling pathways of patulin degradation. Figure 2b,b′,b″ shows the upregulated DEGs according to the GO enrichment analysis at 6 h, 12 h, and 24 h. In the biological process, the cell wall chitin biosynthesis process occurred at 6 h, small-molecule metabolic regulation, small protein removal, and the cellular carbohydrate metabolic process at 12 h, and ribonucleoprotein complex biogenesis and ribosome assembly at 24 h. In terms of cell composition, the DEGs were enriched in the mitochondria-associated ER membrane and cell membrane component. In terms of molecular functions, the DEGs were mainly associated with oxidoreductase, intramolecular transferase, and methyltransferase activities at the initial degradation phase, and were enriched in the acetyl-CoA, coenzyme, organic anion, and amide transmembrane transporter activities at the mid and late degradation phases. The metabolic signaling pathways of the DEGs are shown in Figure 2c,c′,c″; there were 20, 8, and 14 KEGG pathways enriched by DEGs at 6 h, 12 h, and 24 h, respectively. Among them, mitophagy, sulfur metabolism, the biosynthesis of amino acids, and secondary metabolites, were upregulated at the initial degradation phase, while ABC transporters, carbon metabolism, peroxisome, the biosynthesis of secondary metabolites, and autophagy pathways dominated during the main degradation phase. The results were consistent with those of the GO enrichment analysis in that these processes were involved in the patulin degradation mechanism in S15-8. During the initial degradation phase, S15-8 cells protected themselves from oxidative damage by repairing cell membranes and strengthening cell walls under patulin stress; thus, the chitin biosynthetic process, cell membrane component, mitophagy biological processes, and metabolic pathways occurred significantly. Similar to our previous research, the patulin was degraded into the breakdown products, and the metabolites were then transported out of the cells during the main degradation phase, which ranged from 12 h to 24 h; thus, small-molecule metabolic regulation, secondary metabolites, and various related enzyme activities showed significant upregulation accompanied by enhancements in molecular transport and amide transmembrane transporter activities [14].

### 2.3. qRT-PCR Verification

According to the Venn diagrams (Figure 3a), 25, 9, and 19, differentially expressed genes were induced by patulin at 6 h, 12 h, and 24 h, respectively, which may be the key genes associated with patulin metabolism in S15-8. During the induction of the differentially expressed genes, five primary significantly upregulated genes, A5D9S1, A5DE53, A5DNZ9, A5DKH8, and A5DKI9, which were related to organized enzymes such as short-chain dehydrogenase, S-adenosylmethionine-dependent methyltransferase, cell wall synthesis, glutamine amidotransferase, and ATP-binding multidrug transporter, were all measured and verified using qRT-PCR. The relative expression level of the selected genes compared to β-actin was calculated using the 2^−ΔΔCt^ method. The results in Figure 3b,c show the relationship between the gene expressions according to degradation time (RNA-seq) and patulin concentration at 12 h (qRT-PCR). In general, the expression trend of the genes in the qRT-PCR showed good consistency with that in the RNA sequencing (RNA-seq) analysis. Gene A5D9S1, which encodes dehydrogenase, was not only significantly expressed in the RNA-seq during the main degradation phase (Log_2_FC = 6.75), but also exhibited a significant linear relationship with patulin concentration in the qRT-PCR among the tested genes; thus, it was speculated that the protein may be a potential patulin-degrading enzyme. Gene A5DE53 (encoded S-adenosylmethionine dependent methyltransferase) and A5DNZ9 (cell wall synthesis) showed differential expression at 6 h during the initial degradation phase, mainly related to the toxic stress regulation by repairing cell membranes and strengthening cell walls under patulin stress. Gene A5DKH8, A5DKI9-encoded glutamine amidotransferase and ATP-binding multidrug transport were significantly upregulated after 12 h under patulin stress, which indicated that they were involved in main degradation phase of patulin; this was accompanied by an enhancement in molecular transport and amide transmembrane transporter activities, which are mainly related to cell responses to drug transport across membranes and metabolite transportation.

### 2.4. Gene A5D9S1 Characteristics and Regulatory Factor Prediction

To further explicate the potential degrading enzyme, UniProt was used to classify and characterize the patulin degradation protein encoded by the gene A5D9S1. The results showed that A5D9S1, derived from S15-8, known as an NAD(P)-binding protein, was a member of the short-chain dehydrogenase/reductase family (SDR) with 265 amino acids, and, thus, was called PgSDR-A5D9S1 (Figure 4a). Most members of this family possess at least two domains, with the first binding the coenzyme NAD and the second binding the substrate; these determine substrate specificity and contain amino acids involved in catalysis. Additionally, the possible transcription factors predicted from JASPAP, the location, score, and sequence information of the six possible binding sites of YNR063W, HSF1, DAL80, YPR022C, XBP1, and HAL9 were predicted with a relative profile score threshold of 80% (Figure 4b). However, further sequence alignment, binding site score, and transcription factor function analysis indicated that YNR063W and XBP1 did not belong to *P. guilliermondii*, and DAL80 and YPR022C did not show differential expression during transcription under patulin induction according to the transcriptome sequencing data. Hsf1 and Hal9 were finally selected to verify the relationship between gene expression and patulin induction; their binding sites with the promoter of A5D9S1 are shown in Figure 4c. Figure 4d,e show the result of the gene expression from the RNA-seq and patulin induction from the qRT-PCR verification, respectively. It was found that Hsf1 showed a significant correlation either with PgSDR-A5D9S1 expression in RNA-seq or with patulin concentration in qRT-PCR.

The results in Figure 5 show A5D9S1 and Hsf1 expressions with degradation time (RNA-seq), and gene expression with patulin concentration at 12 h (qRT-PCR). The expression trend of Hsf1 showed good consistency with A5D9S1 in RNA-seq with degradation time, with both of them being significantly expressed during the key degradation phase at 12 h (Figure 5a). Moreover, the expressions of A5D9S1 and Hsf1 also showed a significant correlation with patulin induction at different initial concentrations in the qRT-PCR tests. Yeast HSF1 performs an essential function in response to heat shock stress and various chemical stimuli as a response to proteotoxic stress. In addition to the known response to heat stress, HSF1 was shown to be equally effective in regulating transcriptional changes in response to many conditions, including osmotic stress, oxidative stress, glucose starvation, and proteotoxic stress [15,16,17]. Yeast Hsf1 can regulate the aldehyde-based reductases of SDR family members for aldehyde [18,19]. It was also reported to phosphorylate and activate in response to ginger oleoresin stress by accumulating in the nucleus and increasing the expression of heat shock proteins [20]. Based on the results in this study, showing that transcription factor HSF1 and gene A5D9S1 were induced upon patulin stress, we propose a hypothesis that HSF1 is involved in the degradation of patulin by regulating PgSDR-A5D9S1 expression, while the link between patulin, PgSDR-A5D9S1, HSF1, and their potential molecular mechanism needs to be deeply elaborated and explained in further studies.

### 2.5. Patulin Degradation Ability for PgSDR-A5D9S1-Expressed E. coli

Susceptible *E. coli* BL21 (DE3) and *E. coli* BL21 (DE3) strains carrying the recombinant plasmid pET-30a (+)-PgSDR were exposed to different patulin concentrations of 10 mg/L, 50 mg/L, and 100 mg/L, respectively. All plates were observed after 24 h of incubation at 30 °C. According to the results presented in Figure 6a, compared to the no-patulin group, both BL21 (DE3) and pET-30a (+)-PgSDR-A5D9S1/BL21 (DE3) showed a certain tolerance under the patulin concentration of 10 mg/L, and the inhibitory ability of the latter seemed to be stronger. However, both of their growths were significantly inhibited with a higher patulin concentration at 50 mg/L, and those without pET-30a (+)-PgSDR-A5D9S1 were even more inhibited. None of the strains could survive under a patulin concentration of 100 mg/L, showing that, compared with empty BL21 (DE3), the capability of patulin tolerance in pET-30a (+)-PgSDR-A5D9S1/BL21 (DE3) was enhanced to a certain extent. Based on the patulin tolerance study, a further patulin degradation capability study was conducted. The results in Figure 6b show the capabilities of patulin degradation for pET-30a (+)-PgSDR-A5D9S1/BL21 (DE3) and the susceptibility of BL21 and S15-8, respectively, using MES buffer solution as a control. S15-8 and pET-30a (+)-PgSDR-A5D9S1/BL21 (DE3) showed significant patulin degradation ability with an increase in the degradation time compared with empty BL21 (DE3) and MES. Approximately 92% of the patulin was degraded by pET-30a (+)-PgSDR-A5D9S1/BL21 (DE3) at 16 h compared to 71.6% with S15-8 and 11% with the empty vector. The patulin content was completely degraded within 24 h in pET-30a (+)-PgSDR-A5D9S1/BL21 (DE3), demonstrating that, compared with the original strain S15-8, the patulin degradation ability of the target enzyme PgSDR-A5D9S1 was significantly enhanced by the construction of the overexpression vector.

### 2.6. PgSDR-A5D9S1 Structure Prediction and PgSDR-Patulin Docking

The secondary structure of the target protein PgSDR-A5D9S1 was analyzed using SOMPA to deeply reveal the patulin degradation mechanism. The analysis showed the following secondary structure composition: α-helices constituted 40.98%, β-turns were 6.02%, random coils comprised 37.22%, and extended chains constituted 15.79% (Figure 7a). Subsequently, the 3D structure of PgSDR-A5D9S1 was obtained via the homology modeling method in SWISS-MODEL (Figure 7b). The result indicated that the homology between the protein sequence to be modeled and the template sequence was 84.09%, signifying high-quality protein structure prediction, and the GMQE score, QMEANDisCo result, and QMEAN4 value of the model were 0.95, 0.63 ± 0.05, and −0.85, respectively [21]. In addition, the optimized protein models were evaluated with WHATCHECK, ERRAT, and PROCHECK. The percentage of green regions in the WHATCHECK assay was 55.31% and that of yellow regions was 27.66%. The ERRAT displayed the error of non-bonding interactions between different atom types to determine the presence of anomalous conformation, with an overall quality factor of 98.4375, indicating that the constructed PgSDR model was of good quality. The atomic spacing, torsion angle, and hydrogen bonding of the crystallographic model were evaluated using PROCHECK. The results of the generated Ramachandran plots showed that 95.3% of the residues were located in the favorable regions and 4.7% in the permissive regions, suggesting that the protein conformation was reasonable, and the preset 3D modeling of most of the amino acid residues showed acceptable Psi/Phi torsion angles (Figure 7c) [22]. Molecular docking was subsequently carried out after obtaining the protein structural model that met the requirements; the results presented in Figure 7d show that, with an average binding energy of −3.8 kcal/mol for the best conformation, the major binding sites were ALA-257, PHE-264, SER-256, ASN-258, CYS-194, PHE-254, VAL-263, GLY-147, TRP-153, VAL-222, PHE-226, and ASN-223 based on the two-dimensional interactions of patulin in the protein model. Patulin formed hydrogen bonds with the amino acid residues ALA-257, PHE-264, and SER-256 of the protein receptor at distances of 1.82 Å, 2.38 Å, and 2.22 Å, respectively. In addition, the benzene ring portion of the patulin formed an alkyl interaction with ALA-257. The amino acid residues ASN-258, CYS-194, PHE-254, VAL-263, GLY-147, TRP-153, VAL-222, PHE-226, and ASN-223 around the patulin also formed van der Waals force interactions, enhancing the structural stability of the patulin and PgSDR-A5D9S1 complexes.

## 3. Conclusions

In this study, the molecular mechanism of patulin degradation was preliminarily explored. It was found that the changes in significant gene expression were closely associated with different patulin degradation times. The functional and metabolic pathway enrichment analyses of the DEGs revealed key pathways and genes closely related to patulin degradation. The short-chain dehydrogenase PgSDR-A5D9S1 showed significantly high expression in RNA-seq and a significant linear relationship with patulin concentration, presumably making this protein a potential patulin-degrading enzyme. The exogenous expression system of PgSDR-A5D9S1 in *E. coli* was constructed and demonstrated a strong patulin tolerance and degradation ability. This is important for the development of effective patulin-degrading commercial enzyme preparations and the application of patulin detoxification in food. In addition, combining the results of RNA-seq and qRT-PCR validation, heat-excited protein HSF1 was predicted as the transcription factor regulating target protein PgSDR-A5D9S1; this may provide clues to the further analysis of the patulin degradation mechanism. However, detailed analyses of the specific regulatory mechanisms and the target proteins still need to be carried out, and this will be the direction and focus of our future study.

## 4. Materials and Methods

### 4.1. Patulin and Yeast Strain Preparation

Patulin standard, purchased from Sigma-Aldrich (St. Louis, MO, USA), was dissolved in ethyl acetate to prepare a 100 mg/L stock solution, stored at −80 °C. The probiotic yeast strain *Pichia guilliermondii* S15-8 used in this study was provided by the Food Nutrition and Safety Laboratory at the College of Food Engineering and Nutritional Science of Shaanxi Normal University. The yeast strain was cultivated in yeast extract peptone dextrose (YPD, glucose 2%, peptone 2% and yeast extract powder 1%) medium at 120 rpm, 30 °C for 24 h.

### 4.2. Patulin Degradation and Quantification

After one culture cycle of activation incubation, *P. guilliermondii* S15-8 strain was collected to conduct the patulin degradation assay. Yeast cells (1 × 10^7^ cells/mL) were inoculated (with 5% inoculation) into 5 mL YPD medium containing patulin 1.0 mg/mL, placed into the shake cultivation with 120 rpm, 30 °C for 20 h.

After degradation, the supernatants were collected for patulin extraction with 10 mL ethyl acetate three times. Ethyl acetate extracts were collected and distilled with a rotary evaporator at 40 °C. Then, patulin was resuspended with 1 mL deionized water; the samples were analyzed by HPLC after microfiltration. The HPLC analysis was carried out using a mobile phase composed of a mixture of water and acetonitrile (90:10, *v*/*v*) at a flow rate of 1 mL/min, with the detection wavelength set at 276 nm. Patulin quantification and the degradation efficiency calculation were performed as previous studies have described [14,23].

### 4.3. RNA Extraction and Sequencing

After activation incubation, *P. guilliermondii* S15-8 at the concentration of 1 × 10^7^ cells/mL were inoculated (with 5% inoculation amount) into four different reactors with 10 mL YPD medium containing patulin 1.0 mg/L, and placed into the shake cultivation at 120 rpm, 30 °C for 0 h, 6 h, 12 h, and 24 h, respectively. Each treatment group had three biological replicates. The cells were then collected and washed 3 times with ddH_2_O after centrifugation at 3300× *g* for 5 min. Subsequently, the obtained cells in different reaction times were frozen and ground in liquid nitrogen. Total RNA from each sample was extracted with a Yeast RNAiso Kit (TAKARA, Dalian, China). RNA quality was determined and accurately tested. The qualified RNA was used for subsequent Illumina sequencing. RNA-seq profiling was performed by Shenzhen Chengqi Biotechnology Co., Ltd. (Shenzhen, China).

### 4.4. Enrichment Analysis and qRT-PCR Verification

Gene expression differential analysis was performed using DESeq2, and genes with corrected *p* < 0.05 were annotated as differentially expressed, respectively. The R software cluster Profiler package was used, and the screened DEGs were analyzed for GO and KEGG enrichment to identify key DEGs associated with patulin degradation [24,25].

To verify the accuracy of the transcriptome sequencing results, the DEGs were selected for qRT-PCR verification; specific primers were designed using primer 5.0 software. The qRT-PCR reaction system was 4 µL of cDNA template, 0.4 µL each of forward and reverse primers (10 µmol/L), 10 µL of SuperMix, and ddH_2_O supplemented to 20 µL. The reaction program was performed on a CFX96™ Real Time PCR system. The amplification conditions were as follows: 50 °C for 2 min; 95 °C for 10 min; 95 °C for 15 s, 58 °C for 15 s, and 72 °C for 20 s, with 40 cycles. The dissolution curve program used to determine the specificity of the reaction was as follows: 60 °C for 15 s, then 95 °C for 15 s. After amplification, the relative expression of specific genes was calculated according to the 2^−ΔΔCT^ method, and three biological replicates were analyzed for each gene [26].

### 4.5. Target Gene Characteristics and Regulatory Factor Prediction

Multiple-sequence alignment analysis was used to classify the patulin degradation protein encoded by the target gene. The gene sequences of the upstream 3 kb and downstream 1 kb promoter region of target gene were obtained. Then, the possible transcription factors were predicted from the JASPAR database, and a follow-up qRT-PCR verification study was performed to verify the transcription factor of patulin degradation protein.

### 4.6. Target Gene Clone, Plasmid Construction, and Transformation

Firstly, target gene amplification was performed through goal-directed gene amplification, with primers designed for the target gene based on the sequence of *Meyerozyma guilliermondii* ATCC 6260. Subsequently, a recombinant plasmid construction was conducted; the target full-length gene was cloned into a linearized pET-30a (+) expression vector by NdeI/XhoI. The recombinant plasmid was verified and confirmed by colony PCR and plasmid DNA sequencing. Then, the resulting plasmids were transformed into *E. coli* BL21 (DE3) for target protein expression.

### 4.7. Patulin Sensitivity and Degradation Assay in Target Gene Expressed E. coli Strain

After activating the target gene expressed *E. coli* strain, the cultures were diluted 5-fold and 10-fold, and subsequently spotted onto solid LB culture medium containing 0, 10, 50, and 100 µg/mL patulin (3 μL/spot). Each experimental group consisted of three parallel samples. The Petri dishes were incubated at 30 °C for 72 h, and the growth of bacterial colonies was observed.

To investigate the degradation capability of the target gene expressed *E. coli* strain towards patulin, the induced bacterial cultures with IPTG were co-cultured with patulin. Subsequently, the cultures were centrifuged at 3300× *g* for 5 min and subjected to three rounds of sterile distilled water washing. The bacterial cells were resuspended in a 50 mmol/L MES buffer at pH 6.0 to achieve an OD 600 ranging from 0.5 to 0.6. The bacterial suspension containing 2 µg/mL concentration of patulin was then cultivated on a shaker at 37 °C. *P. guilliermondii* S15-8 and susceptible *E. coli* were employed as control groups. Following cultivation, the samples were centrifuged, and the supernatant was collected and filtered through a 0.22 µm filter for subsequent HPLC analysis. Samples at 8 h, 16 h, 24 h, 36 h, and 48 h were analyzed to investigate the dynamic changes in patulin content.

### 4.8. Three-Dimensional (3D) Structure Prediction and Molecular Docking of Patulin Degradation Protein

The 3D structure of patulin degradation protein was predicted using the online website SWISS-MODEL [27]. The secondary structure of the patulin degradation protein was analyzed using SOMPA to deeply reveal the degradation mechanism of patulin. The tridimensional structure of patulin was procured from the ZINC database. Preceding the initiation of molecular docking, both the ligand and receptor underwent protonation and charge assignment. Molecular docking was conducted using the Auto Dock Vina 1.2.2 software, with binding energy calculations utilized to assess molecular binding activity [28,29,30]. The binding mode of patulin to the protein was visualized using Pymol 2.1 and BIOVIA Discovery Studio 4.5 software.

### 4.9. Data Analysis

The experimental data were subjected to statistical analysis using SPSS 26.0, with a significance level set at *p* < 0.05. Origin 2018 software was used for plotting. All the tests were repeated three times.

## Figures and Tables

**Figure 1 toxins-16-00177-f001:**
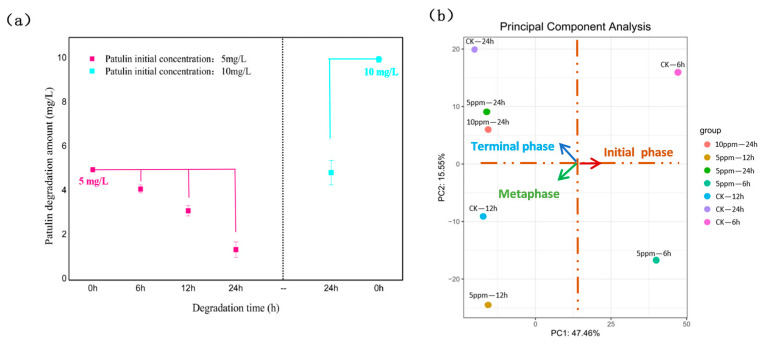
Patulin degradation capacity of S15-8 (**a**), and RNA-seq PCA analysis (**b**) in different degradation times and concentrations.

**Figure 2 toxins-16-00177-f002:**
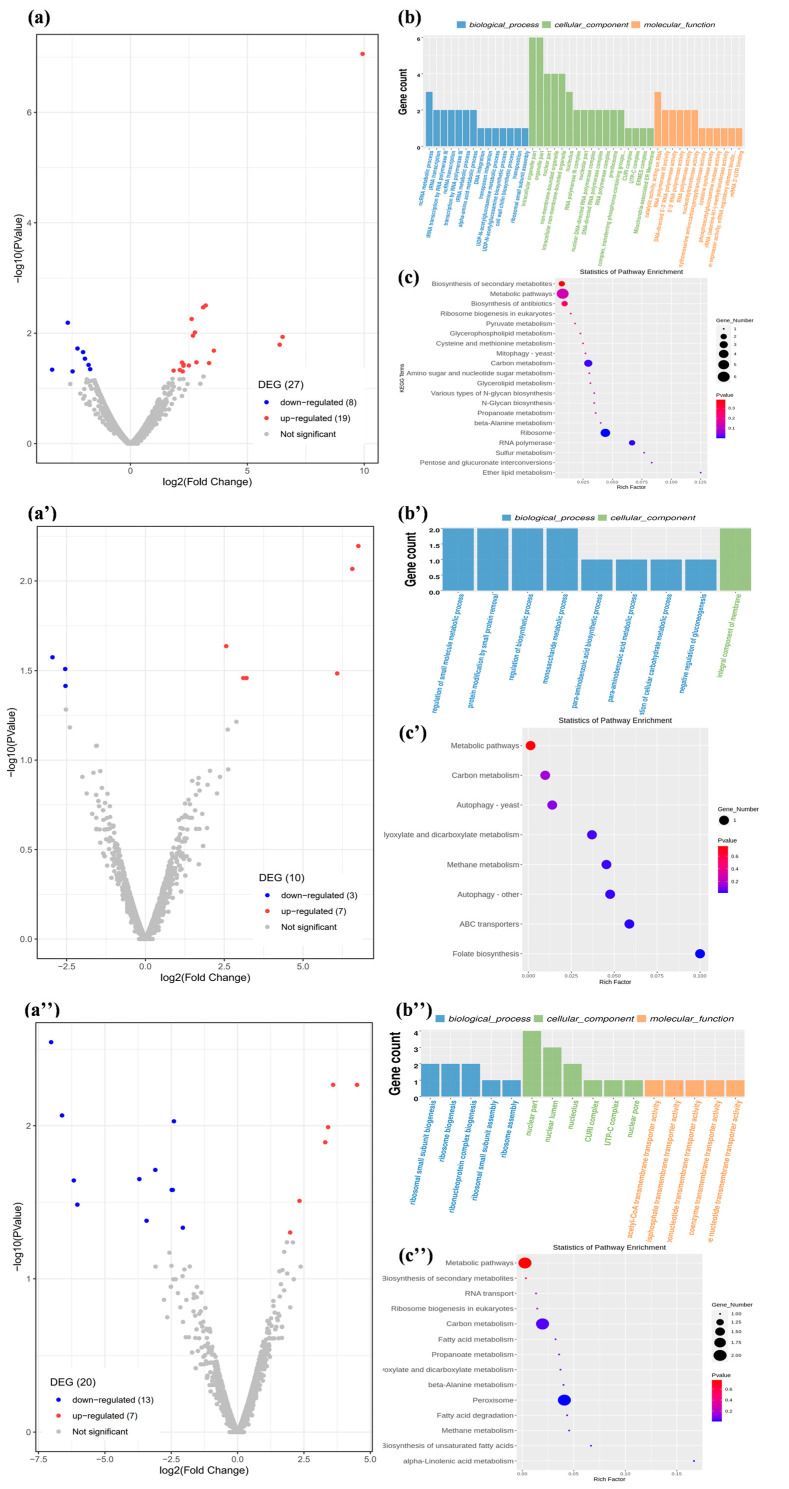
RNA-seq analysis. (**a**,**a′**,**a″**) are the distribution of differentially upregulated and downregulated genes at 6 h, 12 h, and 24 h, respectively; (**b**,**b′**,**b″**) are the GO analysis of the upregulated DEGs at 6 h, 12 h, and 24 h, respectively; (**c**,**c′**,**c″**) are the KEGG pathway enrichment analysis of DEGs at 6 h, 12 h, and 24 h, respectively.

**Figure 3 toxins-16-00177-f003:**
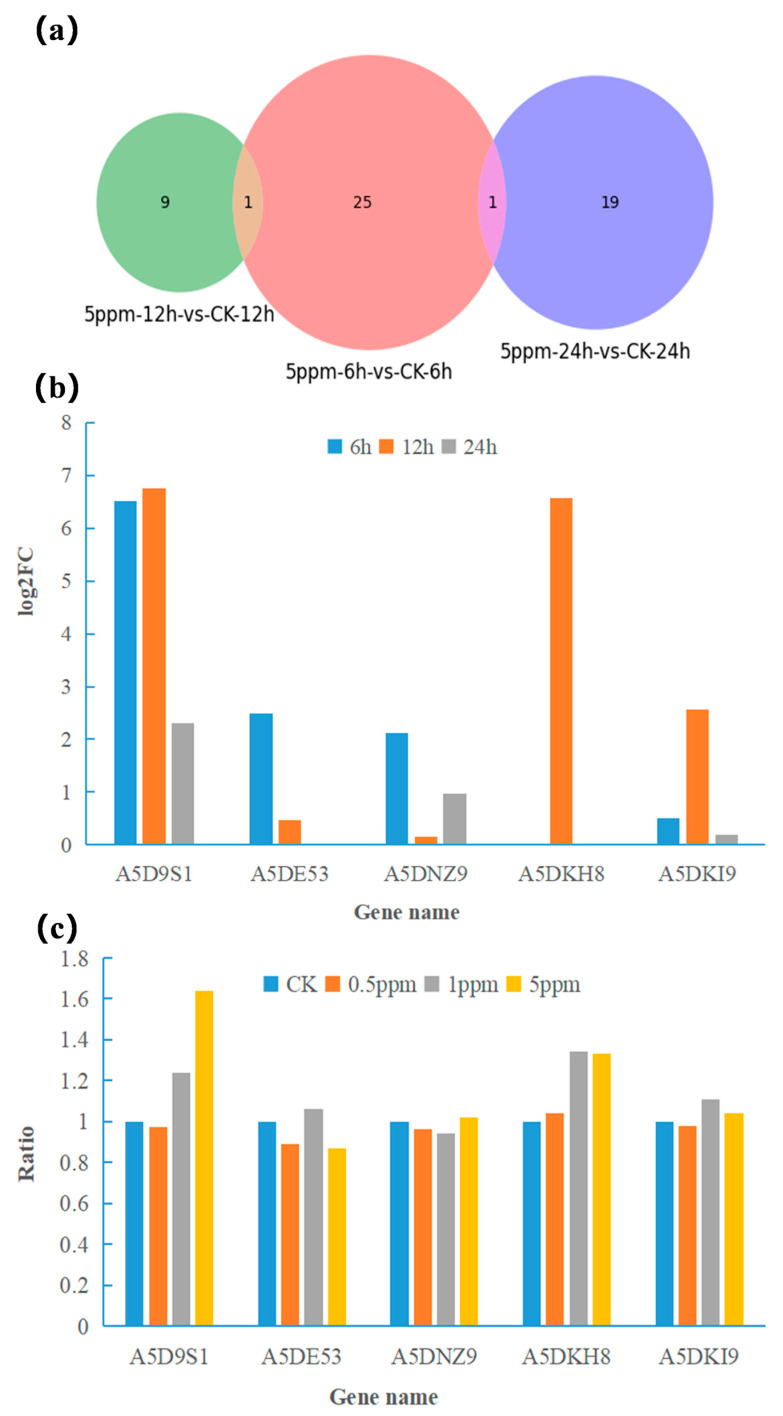
Venn diagrams of transcripts in different degradation times (**a**). (**b**,**c**) are the validation of the five upregulated DEGs with degradation time (RNA-seq) and patulin concentration (qRT-PCR).

**Figure 4 toxins-16-00177-f004:**
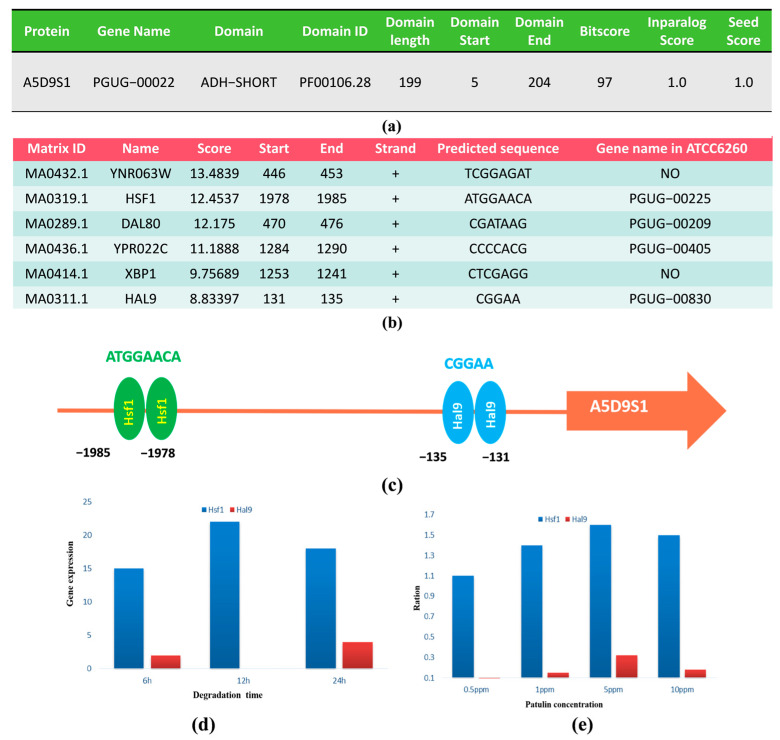
Gene A5D9S1 characteristics and regulatory factor prediction. (**a**) Gene A5D9S1 encodes a protein involved in classification and characterization; (**b**) Possible transcription factors predicted from JASPAP; (**c**) Simulation of binding sites between two potential transcription factors and the promoter of A5D9S1; (**d**,**e**) are the validation of gene expression for potential transcription factors in RNA-seq and patulin-induced conditions.

**Figure 5 toxins-16-00177-f005:**
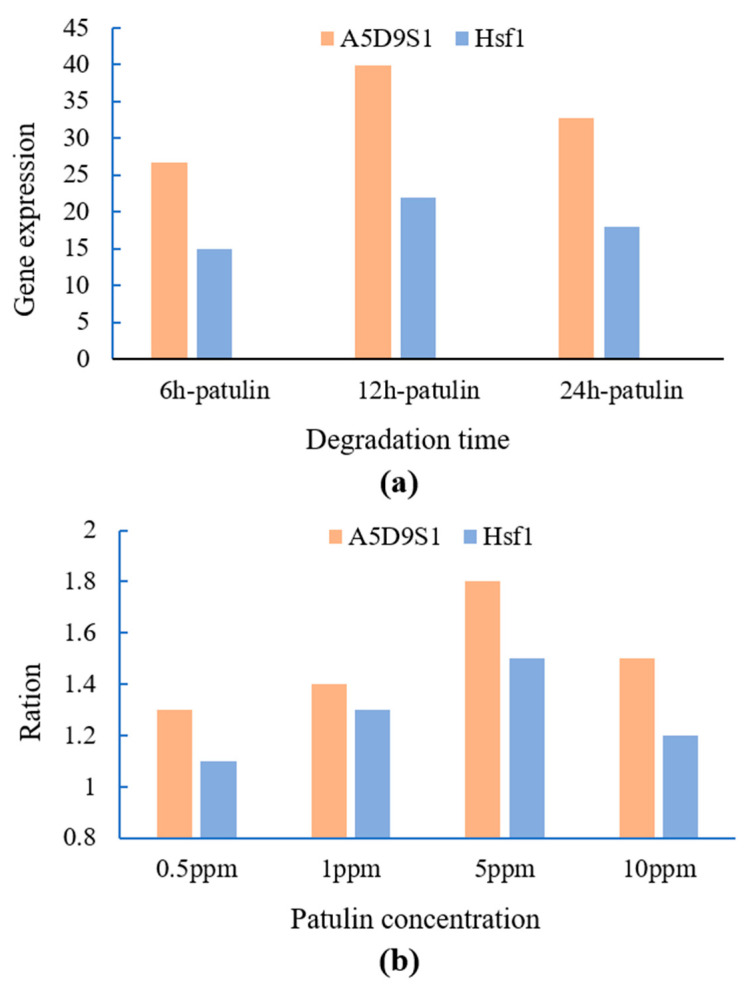
Gene A5D9S1 and Hsf1 expressions with degradation time (RNA-seq) (**a**), and gene Hsf1 expression with patulin concentration (**b**).

**Figure 6 toxins-16-00177-f006:**
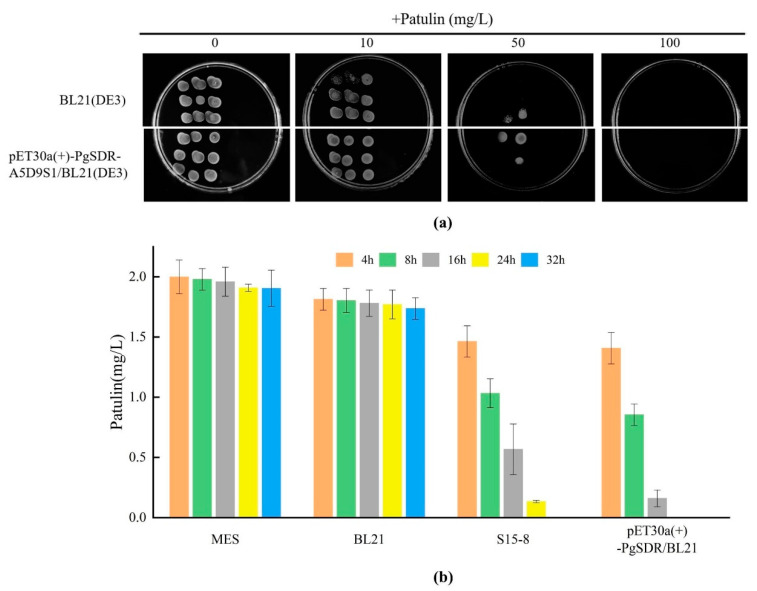
Patulin tolerance and degradation ability of the PgSDR-expressed *E. coli* strain. (**a**) Sensitivity of PgSDR-expressed *E. coli* BL21 (DE3) strain to different concentrations of patulin; (**b**) Patulin degradation ability of the PgSDR-expressed *E. coli* BL21 (DE3).

**Figure 7 toxins-16-00177-f007:**
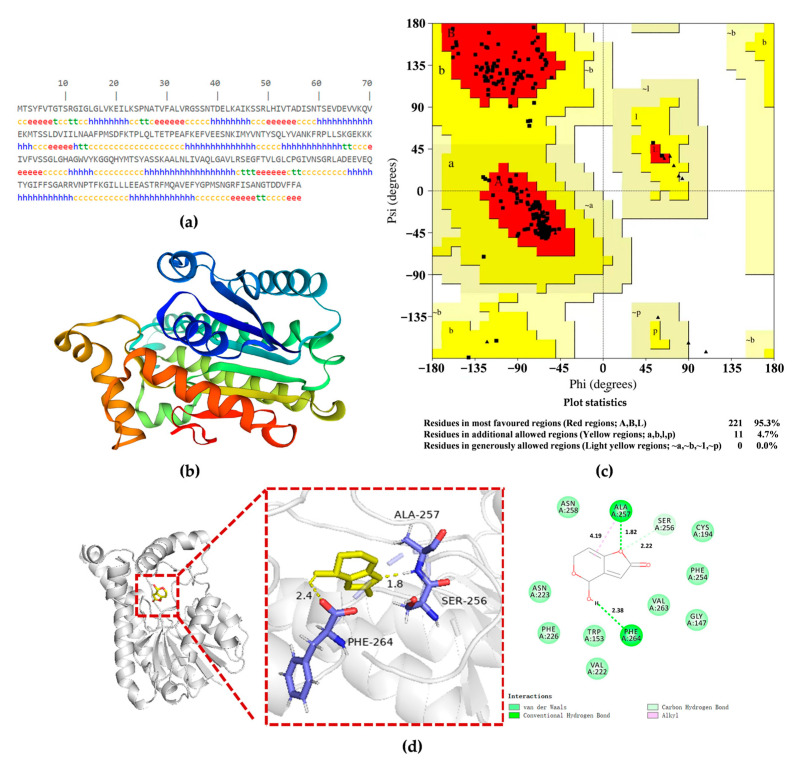
Structure prediction and molecular docking analysis of PgSDR-A5D9S1. (**a**) Secondary structure of PgSDR-A5D9S1 predicted by SOMPA; (**b**) 3D structure of PgSDR-A5D9S1 predicted by SWISS-MODEL; (**c**) Assessment of protein conformational plausibility by Ramachandran plot; (**d**) Molecular docking analysis of the binding capacity of patulin with PgSDR-A5D9S1.

## Data Availability

The data presented in this study are available on request from the corresponding authors.

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
