# Peer review of "Patulin Biodegradation Mechanism Study in Pichia guilliermondii S15-8 Based on PgSDR-A5D9S1"

_toxins, 2024, doi:10.3390/toxins16040177_

Round 1

Reviewer 1 Report

Comments and Suggestions for Authors

The manuscript  described  patulin biodegradation mechanism in a probiotic yeast Pichia guilliermondii S15-8 7. As concluded the authors they proposed "a theoretical 15 basis and technical support for the industrial application of biodegradable functional strains". Method adopted is quite specific and there are a lot of abbreviations that make difficult to read by a not expert. This is a problem because to apply the results of the study professionals with other expertise need to understand the proposed theory.

Unity for centrifugation could be clearer if expressed as xg

The quality of figures should be improved. 

It is need a list of abbreviations.

Author Response

We have carefully considered all of the reviewer's comments and have revised the manuscript accordingly. The specific responses have been attached. We expect the revised version can meet the journal's publication requirements.

Reviewer 2 Report

Comments and Suggestions for Authors

The study investigates a relevant topic: patulin biodegradation, which is important for food safety. The authors employ a multi-pronged approach, including transcriptome sequencing, qRT-PCR, protein structure prediction, and protein expression for functional analysis. The identification of a potential patulin degradation enzyme (PgSDR-A5D9S1) and its putative regulator (HSF1) is a promising finding.

Based on a critical analysis, given below are several drawbacks of the manuscript for its suitability for publication and improvement:

The basic concept of using microbes for patulin degradation is not new. The manuscript would benefit from a clearer explanation of how this study builds upon existing knowledge.

Even though the authors identify differentially expressed genes, a more in-depth analysis of their potential roles in patulin degradation pathways is needed.

Although the authors express PgSDR-A5D9S1 in E. coli, a more robust demonstration of its patulin degradation activity is required. Ideally, this would involve measuring patulin levels before and after incubation with the engineered E. coli strain.

By addressing these weaknesses and incorporating the suggested improvements, the authors can significantly enhance the quality and impact of their manuscript.

Comments on the Quality of English Language

The manuscript requires editing for clarity, grammar, and sentence structure. Ensure consistent verb tense throughout the manuscript. Break down complex sentences into shorter, more readable ones.

Author Response

(The authors gave the same response as above.)

Reviewer 3 Report

Comments and Suggestions for Authors

Although the experimental work was organized well, the langueg must be improved. 

Comments on the Quality of English Language

Due to many language mistakes, this paper must be extensively reviewed by native speaker before any further decisions. The experimental work is scientifically sound and good, but the language must be rechecked. 

Author Response

Response to Reviewer 3 Comments

We sincerely thank the reviewer for careful reading. We studied the comments in the attachment carefully and have made corrections according to the reviewer’ s good comments. and hope that the corrections will meet with approval. Please find the detailed corresponding revisions in the re-submitted files.

In addition, we feel sorry for the substandard language. In order to make the professional and fluency language expression, we used the paid professional, certified language editing service by native English speakers, and hope that the corrections will meet the language requirements of the journal. The detail revisions were marked with yellow highlights in the present revised manuscript.

Once again, thank you very much for your comments and suggestions.

Round 2

Reviewer 2 Report

Comments and Suggestions for Authors

The only remaining issues are minor grammatical errors, which can easily be rectified during the proofreading of the final galley, if the manuscript is accepted.

Comments on the Quality of English Language

The only remaining issues are minor grammatical errors, which can easily be rectified during the proofreading of the final galley, if the manuscript is accepted.